# Subsequent Malignant Neoplasms in Retinoblastoma Survivors

**DOI:** 10.3390/cancers13061200

**Published:** 2021-03-10

**Authors:** Armida W. M. Fabius, Milo van Hoefen Wijsard, Flora E. van Leeuwen, Annette C. Moll

**Affiliations:** 1Department of Ophthalmology, Amsterdam UMC, Vrije Universiteit Amsterdam, Cancer Center Amsterdam, 1081 HV Amsterdam, The Netherlands; m.vanhoefenwijsard@amsterdamumc.nl (M.v.H.W.); a.moll@amsterdamumc.nl (A.C.M.); 2Department of Epidemiology, The Netherlands Cancer Institute, 1066 CX Amsterdam, The Netherlands; f.v.leeuwen@nki.nl

**Keywords:** retinoblastoma, subsequent malignant neoplasms, second primary malignancies, trilateral Rb, heritable Rb, long-term surveillance

## Abstract

**Simple Summary:**

Currently survival from retinoblastoma exceeds 95% in high-income/resource countries. Life expectancy within the heritable retinoblastoma population is mainly threatened by trilateral retinoblastoma in early childhood and subsequent malignant neoplasms throughout life. In this review the risks of specific subsequent malignant neoplasms and trilateral Rb, age at onset and influence of therapy are examined. Furthermore, long-term surveillance guidelines in the heritable retinoblastoma survivors are discussed.

**Abstract:**

Retinoblastoma (Rb) is a pediatric malignant eye tumor. Subsequent malignant neoplasms (SMNs) and trilateral Rb (TRb) are the leading cause of death in heritable Rb patients in developed countries. The high rate of SMNs in heritable Rb patients is attributed to the presence of a mutation in the *RB1* tumor suppressor gene. In addition, Rb therapy choices also influence SMN incidence in this patient group. The incidence rates and age of occurrence for the most frequent SMNs and TRb will be discussed. In addition, the impact of genetic predisposition and Rb treatments on the development of SMNs will be evaluated. Furthermore, screening and other prevention methods will be reviewed.

## 1. SMN Definition and Reason of Susceptibility in Heritable Rb Population

### 1.1. Subsequent Malignant Neoplasms, Trilateral Rb and Metastasis

Subsequent malignant neoplasms (SMNs) after retinoblastoma (Rb) are also denoted as second cancers or second (or subsequent) primary malignancies. These are new tumors with differential histology, which form independently after occurrence of the primary Rb. A combination of genetics and treatment factors, lead to an elevated risk for survivors of heritable Rb to develop SMNs compared to the general population. Several definitions of SMNs after Rb are used; some groups have reported trilateral Rb (TRb) as a SMN [1]. While others, including the authors, consider TRb to be a separate category since these tumors are histologically similar to the primary Rb tumor [2,3]. Examples of SMN sites, with strong evidence of risk are bone and soft tissue (sarcomas), skin (melanomas), brain and spinal cord (CNS tumor). Also carcinomas and hematological malignancies are reported, although the evidence for these SMN sites is respectively moderate and weak [4].

Trilateral Rb is a multifocal Rb where the tumor not only forms in the retina but also in other neuroectodermal regions. These tumors are classified as primitive neuroectodermal tumors (PNETs) and are located in the pineal gland or suprasellar and parasellar regions of the brain.

Metastasis, in contrast to a SMN, is defined as disease disseminated from the primary Rb tumor. Metastatic Rb is very rare in high-income countries, but is the leading cause of death from Rb worldwide [5]. Rb metastasis is most commonly characterized by massive orbital extension in combination with disseminated disease. However, also minimal disseminated disease is seen in Rb patients and can be detected through CRX (conerod homeobox) qPCR on liquid biopsies [6]. Metastatic spread can occur at sites where SMNs also can manifest, for example bone. In case of central nervous system (CNS) metastasis, clustered Rb cells are found in cytospin procedures on spinal fluid [7]. Differences between TRb, SMN and Rb metastasis for risk, site, spread and histology are summarized in Table 1.

### 1.2. Understanding the Relation between Heritable Rb and SMN Development

In 1971 Knudson coined the theory that is now widely known as the two-hit hypothesis [8]. Currently, in the light of retinoblastoma (Rb) this theory entails that when both *RB1* alleles are inactivated (two-hits) in developing retina cells, this consequently is thought to lead to Rb development.

Heritable Rb patients have a germline Rb1 mutation in one allele in their genome, thereby a mutation in the other allele is sufficient to develop Rb (with some somatic mosaicism exceptions). This heritable *RB1* inactivating mutation can be familial but is mostly de novo. In non-heritable patients, both alleles must acquire a mutations in order to develop the disease. It has been estimated that the distribution of heritable versus non-heritable Rb patients is roughly 50–50% [7,9,10]. Very recently, Zou and colleagues confirmed this estimate; 48.3% heritable Rb cases were detected in 149 probands with Rb in a sequencing study (Zou et al., molecular vision, 2021, PMID: 33456302).

Heritable Rb patients most often develop bilateral disease with earlier Rb onset compared to the non-heritable Rb patients. Heritable Rb patients have an increased risk of cancer compared to the general population. The non-heritable Rb population exclusively develops unilateral Rb (and has no elevated risk of SMNs) [11,12].

After the period of retinal development in early childhood, no Rb can develop anymore. Since *RB1* is a tumor suppressor gene, germline inactivation of one *RB1* allele in all cells of heritable Rb survivors make them prone to development of SMNs. Therefore, this population traverses through multiple windows of susceptibility to various but discrete types of SMNs throughout their lives [1]. Non-heritable Rb patients develop Rb through mutations restricted to cells in the retina, and thereby have no elevated risk for SMN [11].

## 2. Rb Treatment over the Past Decades

### 2.1. Rb Survival and Therapies

The increased incidence of specific SMNs is, at least partially, due to increased overall survival of Rb patients and improvement of Rb therapies. To understand the impact of these factors, increased Rb survival and a chronologic overview of Rb therapies are discussed below.

Whereas less than a century ago nearly all children with Rb died of metastatic disease, currently Rb survival exceeds 95% in high-income/resource countries. Consequently, SMNs are now the leading cause of death in patients with heritable Rb [12,13,14]. Long term survival in patients with non-heritable Rb is comparable to the general population, with no increase in cancer related deaths [15].

Principal aims of Rb treatment are: (1) save the life of the child, (2) to retain as much vision as possible, and (3) preservation of the eye. Treatment options include enucleation (removal of the eye), systemic or targeted chemotherapy (intra-arterial or intra-vitreal), focal therapies such as cryotherapy, laser photocoagulation, plaque radiation therapy, and external beam radiation therapy (EBRT). Medical, cultural and economic issues play a role in treatment choice and often a combination of the above mentioned therapies is used.

### 2.2. Rb Treatment in Time and Effects on SMN Development

In recent decades it has become clear how certain therapies influence SMN development. This leads to changes in therapy regimens, which are described below. A more detailed discussion of incidence and mortality can be found in the section titled “Causes of SMN development”.

Enucleation treatments were first performed in the middle of the 19th century, and are considered the first successful Rb therapy [7]. The first treatment with radiotherapy in ocular cancer (presumably Rb) was described by an ophthalmologist in 1903 [16]. From the second half of the 20th century radiotherapy was routinely used. As early as in the 1960s suspicions arose that radiotherapy could induce SMNs in Rb survivors [17,18,19].

In the 1970s, radiotherapy combined with cryotherapy and light coagulation had increased Rb survival with useful vision to around 80% [20]. As early as the late 1970s, non-irradiated heritable Rb survivors appeared to have a high risks of SMNs [21]. This created the dogma that heritable Rb survivors have a genetic SMN predisposition. At that moment, it appeared that the risk of SMN in non-irradiated and irradiated individuals was roughly similar [22].

In the 1980s more evidence was gathered and it was suspected that irradiation could increase SMN risk in heritable Rb survivors [23,24]. Consequently, by mid 1990s, primary Rb treatment had changed to systemic chemotherapy [25]. This resulted in a 2–3% use of radiotherapy in the 2000s compared to around 30% in the 1970s and 1980s [26].

Systemic chemotherapy reduces the tumor bulk, followed by focal treatment, leading to further regression of the tumor. This treatment sequence, however, has a relatively poor performance to rescue or salvage eyes with advanced Rb [27]. Such eyes can often be saved with superselective intra-arterial chemotherapy (IAC) that has been increased in use since 2006 [27]. The concept of local intra-arterial chemotherapy treatment was first pioneered in the 1950s with carotid artery injections [28] and adjuvant selective ophthalmic arterial injection treatment has been used since 1987 [29]. Nowadays, superselective IAC, which directly delivers chemotherapy in the ophthalmic artery, is one of the first line therapies for Rb treatment [30].

Another targeted approach, which is increasingly used, is chemotherapy locally injected in the vitreous. This so called intra-vitreal chemotherapy (IVitC) was first used in the 1990s in Japan [31]. Following a pause, the treatment was reintroduced in 2012 with a new injection protocol [32]. This technique is especially effective for treating Rb eyes that have vitreous seed. Targeted intra-arterial and intra-vitreal chemotherapy are increasingly used in a primary role during the last decade [30,33]. In summary, Rb treatments have changed substantially over time (Figure 1). Consequently, the type and incidence of SMNs in the heritable Rb population are also changing over time.

## 3. Factors That Influence SMN Reporting

Different risk and incidences have been reported in studies on SMNs in Rb patients. Factors such as variation in treatment, different use of SMN definitions, diversity in the Rb patient populations (years of diagnoses, mix between heritable and non-heritable Rb), and duration of follow-up potentially explain the heterogeneity in SMN studies. Furthermore, some SMN studies only include second malignant neoplasms while others also include subsequent third/fourth (and further) malignant neoplasms.

Variations in therapies for primary Rb can affect the incidence and type of SMNs that arise in heritable Rb survivors [7]. Variations in therapies over time (see also Figure 1), socioeconomic status of the country and cultural preferences might influence therapy choices [38] and thereby reports on SMNs. In addition, treatment centers might offer different therapies, depending on availability of treatment options.

Moreover, variable Rb study populations were used in SMN reports, for example there are population-based (i.e., nationwide) and Rb treatment center-based SMN studies. Reports from tertiary cancer treatment centers with a lot of referrals and potentially high risk patients are not directly comparable to population-based Rb reports that study all Rb patients in a specific country. In addition, pediatric Rb cohorts have been studied separately [39]; risk and incidences differ compared to population-based Rb studies due to the early age at onset of certain SMNs in the pediatric population. Furthermore, follow-up time after Rb diagnosis is also critical for reported risk on specific SMNs. Certain SMNs preferentially manifest early in life while others only present significantly more often than in the general population at a much later age.

Another variable that complicates reporting on SMNs is the use of different definitions. For example, different definitions of SMNs, heritable Rb and radiation fields can be found throughout SMN literature. In some SMN reports, TRb is defined as a SMN [1,12] in others TRb is described in a separate category. This complicates comparison of the cumulative incidence of SMNs between cohorts [2,3,9,11,14,39]. In addition, definitions not only differ per center but also per time period. For example, a subsequent cancer in the brain diagnosed 40 years ago might simply have been referred to as brain cancer. In contrast, modern diagnostic techniques will differentiate between TRb (PNET), sarcoma (which often forms in an irradiated field), a noncancerous meningioma or sinonasal Rb which invades the brain (metastasis).

Current literature on SMNs is heterogeneous, hampering comparison of SMN risk across studies (Figure 2). Studies in which these variables are taken into account and harmonized are currently underway.

## 4. Incidence, Risk and Sites of SMNs

### 4.1. Standard Incidence Rate and Absolute Excess Risk

Studies on four large cohort from the United Kingdom [9,40,41], the United States [12,42,43,44,45,46,47], The Netherlands [11,13,48,49,50] and recently Denmark [51] comprising of long-term Rb survivors have provided consistent data regarding the risk of SMNs. These studies are unique as they compare the SMN risk in Rb survivors to the risk of malignancy in the general population within each country. As a measure to report this risk the standardized incidence ratio (SIR) is used. The SIR indicates the relative risk of Rb survivors to develop SMNs compared to the general population corrected for age, sex and calendar year. These studies have in common that the heritable Rb patients are classified by either having bilateral Rb or unilateral Rb with a Rb family history. Unfortunately, these studies have little molecular data for unilateral patients, therefore de novo heritable Rb cannot be excluded in all unilateral Rb patients. The studies differ in their exclusion (Dutch, British and Danish cohort) or inclusion (U.S. cohort) of TRb as SMN. Also, median length of follow-up is variable, respectively 21.9, 25.2 and 24.5 years in the Dutch, U.S. and Danish cohorts, this median was not available for the British cohort. In addition, the period of Rb and SMN diagnoses (Dutch (1945–2007), U.S. (1914–2000), British (1971–2009) and Danish (1914–2014)) varies between the cohorts and this will likely correspond with different treatment modalities in the cohorts.

In all four cohorts, heritable Rb survivors have a highly increased SIR for SMNs compared to the general population. In contrast, the SIR in non-heritable survivors is not significantly elevated compared to the general population in any of the studies. For interpretation of SIRs it is important to realize that tumor incidence in the general population is not zero and that tumors can develop at all age intervals. For the British cohort MacCarthy et al. report a SIR of 13.7 (95% CI [11.3–16.5]) for heritable and 1.5 (95% CI [0.9–2.3]) for non-heritable Rb survivors [9]. Similarly, for the U.S. cohort Kleinerman and colleagues observed a SIR of 19 (95% CI [16–21]) in heritable Rb survivors and a SIR of 1.2 (95% CI [0.7–2.0]) in non-heritable Rb survivors [12]. In addition, a SIR of 20.4 (95% CI [15.6–26.1]) was found for heritable and a SIR of 1.85 (95% CI [0.96–3.24]) for non-heritable Rb survivors in the Dutch cohort [11]. Finally, the Danish cohort reported a SIR of 11.39 (95% CI [7.37–16.38]) in heritable Rb survivors and a SIR of 1.52 (95% CI [0.82–2.60]) for non-heritable survivors. An overview of these observed SIRs, number of patients and observed SMNs in heritable and non-heritable Rb survivors in these four cohorts are listed in Table 2. Indeed, the increased SIRs for heritable Rb survivors is comparable between all four cohort and further supported by the overlap in confidence intervals. The spread in SIRs between the four cohorts is most likely due to factors like time of Rb diagnosis, treatment modalities, length of follow-up, SMN definition (including TRb as SMN in the USA) and population-based (Danish, British and The Netherlands) versus Rb treatment center-based SMN studies (USA). In conclusion, these four studies show that heritable Rb survivors have an 11 to 20 times elevated cancer incidence compared to the general population while non-heritable Rb survivors have no elevated cancer incidence.

As a separate indicator, absolute excess risk (AER) gives more insight into the number of additional SMN cases on top of the number expected in the general population. AER in the heritable Rb survivors ranges between 57.9 (British cohort) [9], 86.1 (Dutch cohort) [11], 97.2 (U.S. cohort) [12] and 70.26 (Danish cohort) [51] per 10,000 survivors per year (Table 2). Furthermore, the 40-year AER of SMNs in heritable Rb survivors increased from 38.8 in the first decade after diagnosis to 261 excess cases per 10,000 survivors per year [11].

The 40 years cumulative incidence for any SMN after Rb diagnosis was 28% (95% CI [21.0–35.0]) and 32.9% (95% CI [27–38.9]) in heritable Rb survivor populations in the Dutch cohort and U.S cohort respectively [11,12]. Similarly, a recent German reference center study also showed a cumulative incidence of 35.2% (95% CI [28.2–43.5]) to develop SMNs 40 years after Rb in heritable survivors [52]. The cumulative incidence at 40 years was not described for the Danish and British cohorts. However, the British cohort study reported a cumulative incidence at end of interval of 68.8% (95% CI [48.0–87.4]) for heritable Rb survivors aged 25–84 years [9]. The Danish cohort study reported a 60 years cumulative incidence of 51% in heritable Rb survivors (significance (*p* < 0.001) compared to non-heritable Rb survivors with hazard ratio 5.0 (95% CI [2.5–10.3])) [51]. Cumulative incidence in a pediatric population was studied in the German pediatric survivor cohort; cumulative incidence was 5.2% (95% CI [1.7–8.7%]) at the age of 10 years [39].

In these long-term follow-up studies the heritable survivor population has most likely been underestimated in the cohorts. This is due to the fact that the classification between heritable and non-heritable Rb in these cohorts is usually not based on genetic profiling but on laterality and familial history. Potentially, part of these non-heritable Rb survivors are misclassified and are actually heritable Rb survivors who carry a de novo germline *RB1* mutation.

### 4.2. Overall Long-Term Survival and Mortality

Heritable Rb survivors also have a significantly increased standardized mortality ratio (SMR) and are 12.8 times (95% CI [9.6–16.5] more likely to die due to SMNs compared to the general population. No significant changes in mortality were found for non-heritable Rb survivors [13].

Heritable Rb survivors with low stage Rb (International Rb staging system (IRSS) stage 0 or I) are still prone to SMNs and survival-rate as a result declines quite quickly with age. This is illustrated by a 97.4% (95% CI (96.0–98.8%)) 5-year versus a 79.5% (95% CI [74.2–84.8%]) 40-year survival rate in heritable patients with IRSS group 0 or I [53].

The cumulative mortality from SMNs in the U.S. cohort 70 years after heritable Rb diagnosis was 75.8% (95% CI [69–83]) and mortality was 8.5 times higher compared to the U.S. general population (SMR 8.5; 95% CI [7.7–9.2]) [15]. Altogether, in heritable Rb survivors the increased risk for SMNs results in SMNs being the main cause of death in this population [15].

### 4.3. Multiple Subsequent Malignant Neoplasms

Patients with a first SMNs who survive after treatment are still at risk for other SMNs [50,54]. The SIR (8.5 (95% CI [3.7–16.7]) of a second SMN is still increased 8-fold compared to the general population with an AER that increased to 202 excess malignant neoplasms per 10,000 person-years. It is important that physicians are aware of this continued risk since this might favor earlier detection of next SMNs.

### 4.4. Incidence of Specific SMNs (and TRb) and Age at Onset

Published risk evidence for different types of SMNs in heritable Rb survivors was recently assessed by a panel of providers from over the world [4]. Heritable Rb survivors have significantly increased risks for sarcoma, melanoma and brain/CNS tumors, the latter only when radiotherapy was received for their Rb treatment. Moderate evidence for increased risk was found for breast cancer (over the age of 40) and lung cancer. For hematological malignancies there is only evidence of risk if the primary Rb has been treated with alkylating chemotherapy, otherwise there are no or low risks for this SMN. Furthermore, no or low evidence of risks have been found for gastrointestinal malignancies (including colon cancer) and thyroid cancer [4]

Previously, Woo and Harbour analyzed published data on SMNs after Rb, calculating combined percentages and age at onset of different SMN types in Rb survivors [1]. Information on Rb diagnosis/treatment was available for 215 of 602 patients and all these Rb survivors in this study were diagnosed and treated for their Rb before the year 2000. Data from Woo and Harbour’s review was adapted; benign tumors were excluded as SMNs and in addition PNETs were excluded, since for this review, trilateral Rb is not considered to be a SMN (Figure 3). The adapted Woo and Harbour data showed that the most common SMN reported subtype in Rb survivors was sarcoma (64%), followed by carcinoma (13%), melanoma (8%), leukemia and lymphoma (4%), CNS tumor (4%) and other SMNs (7%), of the sarcomas SMN subtype 56% were osteosarcomas. (Figure 3a). The median age that the SMN subtypes sarcoma, carcinoma, melanoma, leukemia/lymphoma and CNS tumor were diagnosed is shown in Figure 3b. SMN subtypes tended to occur within specific age ranges, in the first two decades of life sarcomas were predominating. Melanomas occurred mostly in the third decade while carcinomas were predominantly found beyond the third decade [1]. Similarly, high risks were observed for carcinoma (epithelial cancers in lung, bladder, breast) compared to the general population, these cancer types are mostly found later in life and a significant increased risk for these cancers was seen at 30 years of follow-up [11]. TRb occurs very early in life compared to the SMNs in heritable Rb survivors, currently > 95% is diagnosed before the age of 5 years [3] (Figure 3c).

In a German pediatric survivor cohort different latency was seen for solid (e.g., sarcoma, carcinoma, melanoma) versus non-solid SMNs (e.g., leukemia and lymphoma), all solid tumors were seen at an age >5 and all non-solid tumors were diagnosed <5 years of age (TRb excluded as SMN) [39]. In this pediatric cohort the standard incidence risks compared to the general population were highest for sarcoma (SIR = 148.0 (95% CI [39.8–378.9]) followed by leukemia (SIR = 41.4 95% CI [11.1–106.0] [39].

The median age for Rb survivors to develop a SMN is 13 years (in a report were TRb is included as SMN) [1]. However, reported latency depends greatly on duration of follow-up and SMN type [1]. Of note, the reported incidence of the different SMNs and TRb is subject to variables changing over time, and reflects more treatments given in the past than treatments administered recently.

## 5. Causes of SMN Development

### 5.1. Influence of Genetic Predisposition

Germinal mutations in tumor suppressor gene *RB1* pose enhanced risk for SMN development and consequently heritable Rb survivors are at enhanced risk for SMN development compared to non-heritable Rb survivors [21,43,49]. Increased SMN risks compared to the general population were reported in heritable Rb patients who only underwent enucleation and received no additional radiation or chemotherapy [21,43,49]. Within heritable Rb survivors evidence exists that the origin of the *RB1* germline mutation also determines SMN risk. Heritable Rb survivors either have an inherited (also denoted as familial) or de novo (also denoted as sporadic) *RB1* germline mutation. Both types predispose to SMNs, but bilateral Rb survivors with an inherited *RB1* germline mutation have a slightly higher SMN incidence rate [46]. Reasons for this difference might be differential treatment, or treatment at a later age in de novo germline mutation group. However, also increased mosaicism (less bilaterals) in the de novo germline mutation group may play a partial role since group distinction has been based on family history and not on genetic testing in this study.

Possibly, a relationship exists between the phenotypic consequence of the *RB1* mutation and SMN risk in the heritable Rb long-term survivor group [55]. More specifically, increased SMN risk has been observed for nonsense *RB1* mutations (hazard ratio (HR) 3.5 (95% CI [1.8–6.8]) when compared to carriers of an incomplete penetrance *RB1* mutation (HR = 0.2 (95% CI [0.1–0.8])) [55].

Similarly, a more recent study showed that patients heterozygous for regular penetrance *RB1* mutations had a higher risk to develop SMNs compared to patients with incomplete penetrance *RB1* mutations [56]. However, a study on part of a French cohort treated with irradiation did not show any association between regular or incomplete *RB1* penetrance mutations and SMN risk [57].

Taken together, a very clear increased SMN risk in the heritable Rb survivors is seen compared to the general population [21,43,49]. However, more studies are needed to further investigate SMN risk differences between de novo versus inherited *RB1* germline and regular versus incomplete penetrance mutation carriers.

### 5.2. Influence of Radiotherapy

External beam radiation therapy (EBRT) used to be a standard Rb therapy until it became clear that this therapy significantly increased SMN risk while also affecting the eye and anatomy of the orbit itself. Currently, EBRT is mostly avoided due to fear of SMNs and established treatment alternatives such as IAC and IVitC.

A dose-response relationship has been found between dose of external beam irradiation and the risk for osteosarcoma and soft tissue sarcoma in heritable Rb survivors [23,44]. In addition, irradiated Rb survivors have significantly increased SMN risk and earlier age at onset of SMNs compared to non-irradiated Rb survivors. Sarcomas are significantly more often found inside the radiation field while melanomas occurred significantly more often outside the radiation field or in Rb survivors who have not received irradiation therapy [1].

A significant increase in SMN risk has been reported for heritable Rb patients who were treated before the age of 1 with radiation therapy compared to after the age of 1 year [14,58]. However, two recent retrospective studies did not see this correlation [52,57]. Discrepancies in the studies above could be explained by many factors (see Section 3 “factors that influence SMN reporting”), for example the definition of radiation field in these studies could play a role.

The risk for SMNs increased with 3.1-fold (95% CI [2.0–5.3]) in heritable Rb survivors treated with radiotherapy compared to heritable Rb survivors not treated with radiotherapy in the U.S. Rb survivor cohort [12]. Similarly, the observed hazard ratio for this SMN risk after radiation in heritable Rb survivors was 2.8 (95% CI [1.3–6.2] in the Dutch cohort [11]. Consequently, the cumulative long-term incidence of SMNs is significantly higher in irradiated versus non-irradiated heritable Rb survivors [11,12]. For the Dutch cohort the 40-year cumulative incidence for developing a SMN was 33.2% (95% CI [24.6–42.8]) in Rb survivors treated with radiotherapy and 13.3% (95% CI [3.28–23.3]) in non-irradiated heritable Rb survivors [11]. Altogether, in different cohorts an about 3-fold increased risk of SMNs is seen in irradiated heritable Rb survivors compared to non-irradiated heritable Rb survivors [11,52].

Long-term risk for bone and soft-tissue sarcoma after radiation in heritable Rb survivors is highly elevated as earlier discussed. A recent paper from Kleinerman and colleagues provides more in depth guidance of the risks (age, location and sex) of sarcomas which have developed in a Rb population which was diagnosed and treated between 1914 and 2006 [59]. In the heritable Rb population, risks for sarcoma in the head/neck regions are over 10 times higher than for the body/extremities regions. In more detail, much higher risks were seen in the head/neck regions vs. the body/extremities regions for bone sarcoma (SIR 2213 (95% CI [1671–2873]) vs. SIR = 169 (95% CI [115–239])) and soft tissue sarcoma (SIR = 542 (95% CI [418–692]) vs. SIR = 46 (95% CI [31–65])). Bone sarcoma was mostly seen in the adolescent years in head/neck as well as body/extremities regions. However, soft tissue sarcomas in body/extremities regions were rare in early years and rose steeply after 30 years of age, while for head/neck region the peak was in the adolescent years [59].

### 5.3. Influence of Systemic Chemotherapy

Cancer treatment with systemic chemotherapy is known to increase the risk of SMNs in different patient populations other than solely Rb survivors. For example, breast cancer risk is elevated in non-irradiated childhood survivors compared to the general population and this risk is further increased in a dose dependent matter when survivors were treated with antracyclines or alkylating chemotherapy drugs earlier in life (Henderson JCO 2016). Moreover, the risk of subsequent malignant osteosarcomas has been reported to rise linearly with increasing doses of alkylating agents in a cohort of childhood cancer survivors [23,60]. Similarly, anthracyclines and alkylating chemotherapy agents have also been shown to increase the risk of subsequent malignant sarcomas in childhood survivors [61,62]. In the Rb population, the risk of systemic chemotherapy has mostly been shown in studies where a combination regimen of radiation and systemic chemotherapy has been used. After correction for radiation exposure the systemic chemotherapy remained an independent factor that contributed to increased SMN risk.

Chemotherapy (together with radiotherapy) for Rb patients was first used in 1953 but chemotherapy was only widely adopted as primary therapy in the mid 1990s when it was apparent that radiotherapy led to a steep increase of SMNs in Rb survivors. As a result there is little long-term information about the contribution of systemic chemotherapy to the risk of SMNs in heritable Rb survivors. For a subgroup in the U.S. cohort chemotherapy treatment was identified as relative risk factor in a multivariate analysis in long-term Rb survivors with bone cancer in the U.S. Rb survivor cohort [46]. Also, subsequent malignant leukemia is a well-known side effect of systemic chemotherapy in pediatric and adult oncology [63,64]. Gombos and colleagues performed a retrospective case-series review of Rb patients who have developed SMNs and reported that there might be an increased risk for acute myelogenous leukemia (AML), an otherwise very rare SMN in long-term Rb survivors, in chemotherapy treated individuals compared to non-chemotherapy treated individuals [65]. In addition, excess leukemia cases were found in chemotherapy-treated long-term Rb survivors in different cohorts [53,66]. One study has suggested that chemoreduction with vincristine, etoposide and carboplatin (VEC) may not lead to more leukemia cases, however mean follow-up was only 6.7 years in heritable survivors [67].

Most SMN data involving chemotherapy in long-term Rb survivors is from patients who have been treated with radiotherapy plus chemotherapy for Rb. Wong and colleagues have been the first to show that radiotherapy plus an alkylating chemotherapy regimen significantly increased the risks for leiomyosarcoma and bone tumors compared to radiotherapy alone in a long-term Rb survivor cohort [47].

In addition, recent studies in the German cohort showed that long-term overall survival decreased significantly due to SMNs in heritable Rb survivors who had been treated with radiotherapy compared to enucleation or focal therapy and that this effect was aggravated by chemotherapy [53]. Chemotherapy plus radiotherapy increased the cumulative incidence ratio (1000 person years) almost twofold compared to radiotherapy alone (17.7 [12.7–24.1] vs. 9.0 [6.5–12.1]; *p <* 0.005) and chemotherapy was the strongest predictor of SMNs outside the radiotherapy field. However, no significant effect of chemotherapy alone was found, the cumulative SMN incidence ratio was similar to that of patients treated only with enucleation/focal treatments. The authors noted that this may possibly be due to small group size and relative short follow-up in this chemotherapy-alone group compared to the other groups [52].

In a Japanese cohort, the cancer-free 10-year survival rates for non-irradiated Rb survivors were significantly higher in the bilateral population treated with neoadjuvant chemotherapy (*n* = 15) versus the non-neoadjuvant chemotherapy group (*n* = 23). The group described is small but the reason for the significant difference cannot be explained so far. Anyhow, no significant differences were seen between these above described groups for overall 10-year survival [68].

Altogether, low patient numbers and relatively short follow-up hamper data concerning SMN risk in long-term Rb survivors treated with chemotherapy as primary therapy. Increased patient numbers can be studied when different long-term Rb survivor cohorts would be ‘pooled’ together, enabling detection of possibly smaller effects of primary systemic chemotherapy in Rb long-term survivors.

### 5.4. Influence of Targeted Chemotherapy

The Japanese Rb survivor cohort (*n* = 343 and mean follow-up of 6.6 year) has been treated with a mix of different modalities such as radiation, systemic chemotherapy and local chemotherapy. It has been shown that local chemotherapy (including ophthalmic arterial injections) did not increase the rate of SMNs [29]. The Japanese data gives reassurance as their longest follow-up was 21 years but more research with longer follow-up (for more of the heritable Rb survivors in the cohort) is needed in order to definitely determine the effect of local chemotherapy on development of SMNs [27]. For the contemporary targeted chemotherapy IAC [27] similar SMN rates have been reported in the first 10 years of use compared to earlier published SMN rates [69]. However, targeted chemotherapy such as superselective IAC and IVitC have only been used on a large scale for the last decade and therefore long-term data is not yet available.

## 6. Trilateral Rb

### 6.1. Characteristics and Treatment of Trilateral Rb

Median age of TRb manifestation is 26 months [3] and since 1995 >95% is diagnosed before the age of 5 years (Figure 3c) [3]. The timing of intracranial midline PNETs is either synchronous or metasynchronous with Rb diagnosis. In a meta-analysis the Jong and colleagues reported that over 50% of TRb diagnoses are currently made synchronously. The synchronous diagnoses are mostly non-pineal PNETs and PNET size is significantly larger at diagnosis in non-pineal compared to pineal patients (*p* = 0.0012). TRb is sometimes diagnosed due to clinical symptoms, such as intracranial hypertension, which has a poorer prognosis than non-symptomatic TRb. TRb should preferentially be treated with a high dose regimen systemic chemotherapy with autologous stem cell rescue. In addition, surgery or EBRT may serve useful as consolidation or adjuvant treatment for TRb [3].

### 6.2. TRb Incidence and Effect of Ocular Rb Therapy

TRb incidence rates in heritable Rb patients historically range from 5–15% [70]. The reported TRb incidence could be less reliable due to (publication-) bias from relatively small patient groups in some studies. Reports from small patients groups might be unreliable in this context since publication is often specifically done after encountering at least one TRb in a patient group. Therefore, a recent meta-analysis excluded studies with less than 100 patients and showed an adjusted trilateral incidence rate in heritable Rb patients of 3.5% (95% CI [1.2–6.7]) [3].

Heritable Rb patients who have received radiation before the age of 1 year are more prone to development of TRb. However, it is not clear if this is due to genetic predisposition or radiation scatter [14,58]. Furthermore, it is still debated whether previous systemic chemotherapy is protective against trilateral Rb [3]. To date, no paper has convincingly demonstrated that systemic chemotherapy prevents TRb [71].

For targeted chemotherapy only limited data are available; at 5 year follow-up the TRb incidence was 2.7% in a cohort treated with IAC [69], which is similar to the adjusted general TRb incidence of 3.5% (95% CI [1.2–6.7]) recently reported in a meta-analysis [3].

### 6.3. Mortality Due to TRb

Trilateral Rb used to be nearly always fatal [72,73]. However, recent publications have reported successful TRb treatments [74,75,76]. Accordingly, a meta-analysis showed almost no survival of trilateral Rb patients before 1995, while nowadays about half of patients survive long-term. When patients die of TRb, the mean age at death is significantly lower for non-pineal compared to pineal TRb patients (*p* > 0.0001) [3]. Survival chances for both pineal and non-pineal trilateral patients strongly depend on tumor size and patients without leptomeningeal spread do the best [3].

## 7. Long-Term Surveillance in Heritable Rb Patients

### 7.1. SMN Long-Term Surveillance

Heritable Rb survivors have life-long elevated risks for SMNs [3,9,26,47]. In fact, survivors who have received radiation therapy for treatment of their Rb, have a 50% SMN risk during 50 years of follow-up [9].

A recommendation guideline for long-term surveillance in heritable Rb survivors has been published in 2020 [4]. The panel first assessed the risk evidence for different types of SMNs (Figure 4a), the strongest risk evidence was seen for the sarcoma and melanoma. Also for brain and other CNS tumors strong evidence exist but especially when heritable Rb survivors were treated with radiotherapy.

Sadly, at this moment there is no evidence yet that any surveillance method has been able to extend life in heritable Rb survivors. In the United States cohort, melanoma was detected in a late stage, possibly due to decreased visual acuity in this population. Therefore, it is strongly recommended that the Rb heritable survivors are educated on skin protection measures and have an annual skin exam [4]. Moreover, an annual visit to a long term follow-up clinic is recommended in order to ensure further patient education on prompt evaluation of potential SMN signs [10], such as for example persistent sinusitis, pain and skeletal tenderness. The local national cancer surveillance screening programs for other cancers are also recommended for Rb heritable survivors. When possible, surveillance methods without ionizing radiation should be preferred, although there is lack of evidence for this surveillance recommendation. Screening for leiomyosarcoma should not be performed, since it has not been proven to be beneficial and may do harm. Additional surveillance methods beyond recommendation per local guideline are not recommended, since risks are uncertain and/or benefit cannot be anticipated. Also, annual thyroid ultrasound should not be performed in the heritable Rb survivor population since there is no clear increased risk in this population for thyroid cancer [4]. The adapted surveillance recommendations are summarised in Figure 4b.

The use of whole-body (WB) MRI as non-invasive screening modality in long-term follow-up of heritable Rb survivors has been studied in a tertiary cancer center in the USA [77,78]. Forty-seven heritable Rb survivors received annual WB-MRIs and sensitivity and specificity for SMN detection with WB-MRI in this cohort were only 66.7% (95% CI [14.2–96.0]) and 89.7% (95% CI [83.6–93.7]) respectively. In this center, only limited utility of WB-MRI as surveillance modality in these heritable Rb survivors could be shown [78]. A multicenter study should be done in order to assess the impact of WB-MRI screening on SMN-related mortality, however the researchers underlined that also other screening modalities should be developed [78]. Efforts to develop alternative screening modalities for heritable Rb survivors are currently explored in a collaborative study between research centers in France, the Netherlands and Germany [79]. Hopefully this effort will facilitate a blood based screening modality in the future.

### 7.2. TRb Long-Term Surveillance

For TRb screening a baseline brain MRI is standard of care at time of intraocular Rb diagnosis to detect synchronous TRb [80]. Currently, no strong evidence exists that a brain MRI program gives enough benefits; advantages and disadvantages are still debated in literature. It is still questioned if additional brain MRIs should be performed during follow-up to detect metachronous TRb as 98% of TRb is synchronous. Consequently, 311 MRI scans would need to be done in order to detect 1 asynchronous TRb patient and 776 scans would have to be done to save a single life [80]. In addition, overall survival was not increased due to MRI for TRb screening over time in a tertiary cancer center in the USA. However, false negatives did require additional subsequent MRIs under anesthesia [81]. Additional cost-benefit analysis should be performed in order to determine the usefulness of subsequent brain MRI screening is useful in TRb detection [80]. If a brain MRI program would be opted for, than evidence exists that the screening program would only be required until the age of 36 or 40 months and no longer. Moreover, screening should be independent of age of intraocular Rb diagnosis and no specific age interval exist that would require a screening approach with less or more frequent screening within a certain age interval [80].

Taken together, TRb brain MRI screening at the moment of Rb diagnosis is proven effective, but subsequent brain MRI screening program for asynchronous TRb detection is not. For SMN detection in heritable survivors the most important aspect of screening starts by educating the patient, family and (primary) doctors with regard to the increased risk and common signs of SMNs and an action plan in case concerns arise [10].

## 8. Conclusions

Currently survival from Rb exceeds 95% in high-income/resource countries and life expectancy within the heritable Rb population is mainly threatened by TRb in early childhood and SMNs throughout life [12,13,14,15].

### 8.1. TRb

Adjusted TRb incidence is 3.5% in the heritable Rb population [3] and since 1995 >95% of the TRb patients are diagnosed before the age of 5 years. Nowadays about half of the TRb patients survive the disease [3]. Introduction of MRI at Rb diagnosis increased the detection rate of TRb and has become standard of care in many countries. The majority of TRbs manifest synchronously with intraocular Rb. The benefit of additional periodic MRIs in Rb patients is currently debated, a lot of MRIs are needed to detect a single asynchronous TRb [80].

### 8.2. SMN

Heritable Rb survivors have life-long elevated SMN risk compared to the cancers in the general population, their 40-year cumulative SMN incidence is around 30% [11,12,52]. Examples of SMN sites in heritable Rb survivors are bone and soft tissue (sarcomas), skin (melanomas) and brain/CNS tumors [4]. Certain SMNs (sarcomas) manifest early (mostly in first two decades) while other SMNs (carcinoma) predominantly develop later in life. Therapy for primary Rb and also genetic predisposition (de novo vs. inherited Rb) within the heritable Rb population influence SMN risks [25,26,46,47]. Currently, radiotherapy is avoided as much as possible in order to prevent SMNs and it is pivotal that risks of other currently used therapies are assessed as soon as possible.

To better understand how existing and novel therapies influence SMN risk, large groups of Rb survivors should be investigated. In our opinion, the way forward is by ‘pooling’ clinical data from different cohorts in order to identify potential smaller effects of current therapies. Such international collaborative studies are ongoing. A ‘pooled’ study approach will also harmonize pleiotropic factors that hamper comparison of current Rb cohort studies. In addition, this study will produce detailed SMN risk assessment taking into account genetic predisposition and primary Rb therapy. Potentially, these risk assessments can be used to create individual surveillance programs for heritable Rb survivors.

Increased detection of SMNs in heritable Rb survivors is aided by creating awareness in this group (and their doctors). Further improvements in SMN detection could be attained by annual follow-up visits to specialized clinics and prompt evaluation of potential SMN symptoms. Programs that focus on early SMN detection have not yet shown to extend life of heritable Rb survivors. SMN detection results using annual WB-MRIs are underwhelming and multicenter SMN-mortality focused studies are needed in order to show efficacy of WB-MRI in heritable Rb survivors [78]. In the meantime, other screening modalities, such as blood-based screening of heritable Rb survivors, are also explored. Hopefully, these efforts will lead to functional early detection methods that prevent SMN mortality in heritable Rb survivors in the future.

## Figures and Tables

**Figure 1 cancers-13-01200-f001:**
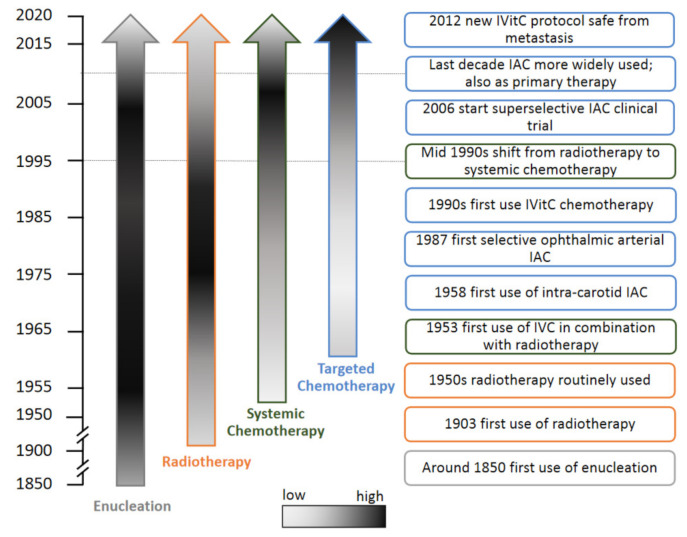
Use of specific Rb therapies over calendar time. Rb therapy changed over time. The relative use of specific Rb therapies (enucleation, radiotherapy, systemic chemotherapy and targeted chemotherapy) over calendar time is shown [7,16,27,29,30,31,32,34,35,36,37]. The scale bar indicates relative use (light grey = low and black=high). IVitC = intra-vitreal chemotherapy, IAC = intra-arterial chemotherapy, IVC = intravenous chemotherapy.

**Figure 2 cancers-13-01200-f002:**
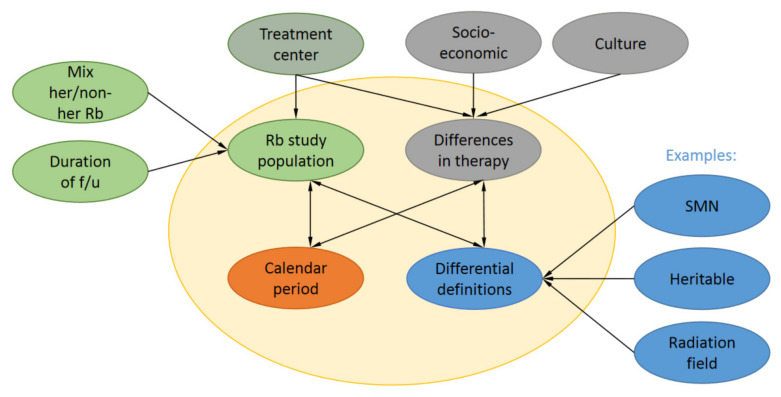
Factors that influence SMN reporting. A range of factors such as Rb study population, calendar period of Rb diagnosis, differential definitions for the same concepts and differences in therapy influence reported risks and incidences in SMN reports. Rb study populations have differential follow-up (f/u) can be either population-based or Rb treatment center-based with a different mix of heritable (her) and non-her Rb survivors. Also, different treatment centers might have slight preferences for certain therapies. Differences in therapy are also influenced by socioeconomic status in the country of treatment and cultural preferences of patients.

**Figure 3 cancers-13-01200-f003:**
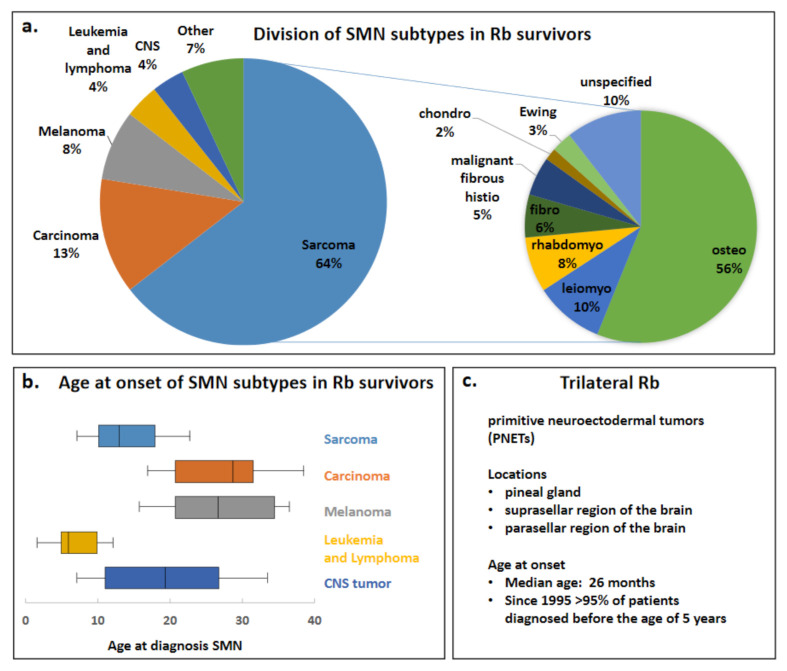
Division of SMN subtypes in Rb survivors and age at onset of SMNs and TRb. Data from the Woo and Harbour review has been adapted; (benign tumors and PNETs have been excluded) to create the pie charts and box plots for depicting the division of specific SMNs in Rb survivors. (**a**) Pie charts of the different SMN types (left) and the sarcoma SMN subtype (right) are shown. (**b**) Box plots of age at onset of diagnosis of SMN subtypes sarcoma, carcinoma, melanoma, leukemia/lymphoma and CNS tumors are shown. The boxes depict the interquartile range and the middle line represents the median. Furthermore, a line extends from the 10th to 90th percentile of values; excluding “outside” values. CNS = central nervous system. (**c**) Trilateral Rb locations and age at onset are adapted from de Jong and colleagues [3].

**Figure 4 cancers-13-01200-f004:**
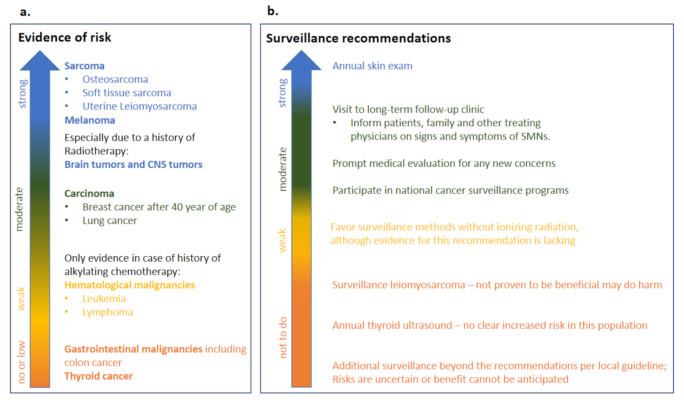
Long-term follow-up in heritable Rb survivors. A panel of providers from over the world recently assessed risk evidence and surveillance recommendations for heritable Rb [4]. This figure shows an adapted Table 1 from Tonorezos and colleagues [4] with (**a**) risk evidence (**b**) surveillance recommendations.

**Table 1 cancers-13-01200-t001:** Differences between SMN, TRb and Rb metastasis.

Sort	SMN	TRb	Rb Metastasis
Increased risk in heritable Rb patients *?	Yes	Yes	Not known, lack of evidence
Spread from Rb?	No	No	Yes
Histology	Different from Rb	Similar to Rb	CNS cytospin may show clustered Rb cells
Site	bone or soft tissue (sarcomas) Skin (melanoma) Brain and spinal cord (CNS tumor) e.g., breast and lung (carcinomas)	Intracranial in neuroectodermal regions of the brain	Bone
			CNS
			CSF
			Bone marrow
			Lymph nodes
			Liver (rare)

Differences between SMN, TRb and Rb metastasis for risk, site, spread and histology are listed [3,4,7]. * compared to non-heritable Rb patients and the general population, CNS = central nervous system, CSF = cerebrospinal fluid.

**Table 2 cancers-13-01200-t002:** SMN risk by heritability in four long-term Rb survivor cohorts.

Group		Heritable Rb Survivors					Non-Heritable Rb Survivors		
	Total No. of Patients	No. of Patients	No. of SMNs	SMN SIR (95% CI)	AER	Cumulative Incidence (95% CI)	No. of Patients	No. of SMNs	SMN SIR (95% CI)
Dutch cohort	668	298	62	20.4 (15.6–26.1)	86.1	at 40 years: 28% (21.0–35.0)	370	12	1.85 (0.96–3.24)
U.S. cohort	1601	963	260	19 (16–21)	97.2	at 40 years: 32.9% (27–38.9)	638	17	1.2 (0.7–2.0)
British cohort	1927	806	146	13.7 (11.3–16.5)	57.9	at end of interval in population aged 25–84 years: 68.8% (48.0–87.4)	1121	23	1.5 (0.9–2.3)
Danish cohort	323	133	25	11.39 (7.37–16.81)	70.26	at 60 years: 51% in heritable Rb (CI is NA)	190	13	1.52 (0.81–2.60)

Overview of SMN risk for heritable and non-heritable Rb survivors in the Dutch [11], U.S. [12], British [9,40] and Danish [51] Rb cohorts. No = number, CI = confidence interval, SIR = standard incidence ratio (observed/expected number; in grey are non-significant SIRs depicted), AER (absolute excess risk) per 10,000 person-years and NA = not available. Decimals places used for numbers in this table correspond to the respective original publications.

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
