# Peer review of "Subsequent Malignant Neoplasms in Retinoblastoma Survivors"

_cancers, 2021, doi:10.3390/cancers13061200_

Round 1

Reviewer 1 Report

In this review article, the authors discussed subsequent primary malignancies (SPMs) with trilateral retinoblastoma (TRb) in patients with retinoblastoma. Major and minor concerns are as follows.

Major concerns:

  1. It is not clear to understand the first part of the manuscript regarding the definition of SPMs, TRb, and metastasis. If the authors think that TRb is different from SPMs, please do not use the expression that TRb is 'a separate type of SPM' in the abstract.
  2. The cellular origin of retinoblastoma: In Line 47, the author mentioned that RB1 alleles are inactivated in cone precursor cells. However, the cellular origin of retinoblastoma is yet to be determined. There are still debates.
  3. The proportion of heritable vs. non-heritable retinoblastoma: It is not roughly half (in Line 49).
  4. Please define the definition of 'standardized incidence ratio' and 'absolute excess risk' in the manuscript (line 188 and 217).
  5. The composition of the manuscript: In the current form, it is not clear what the authors want to emphasize in the manuscript. 
    1) Sites, risks, and incidence of SPMs should be more clearly described in the manuscript.
    2) The flow of the manuscript is difficult to follow. <Rb treatment over the past decades> and <factors influencing SPM reports> should be separated. 
  6. Please include the last section for summary and/or future directions. The current manuscript seems to end abruptly.

Minor concerns:

  1. Please check typographical and grammatical errors throughout the manuscript. One of the errors is the use of 'Retinoblastoma', not 'retinoblastoma' in the middle of sentences. And if the authors decided to use 'Rb' for 'retinoblastoma', please use the abbreviation throughout the manuscript.
  2. How about mentioning the examples of sites of SPMs in the first paragraph? 
  3. In Table 1, please change the order to SPM, TRb, and metastasis, mentioning SPM first.
  4. In Figure 1, what is the meaning of 'frequency' in the scale? Please indicate in the legend.

Reviewer 2 Report

This review on subsequent malignancies in retinoblastomas is comprehensive and summarizes the literature on the subject. The graphics are attractive and helpful. Parts of the review do a good job of summarizing and interpreting the research; other parts read more like a list of results with little interpretation to help the reader understand the "big picture."

The section on Factors that influence SPM reporting starting on page 4 was excellent including the final paragraph that the studies are heterogenous, which hampers comparison among studies. I would have liked to see this heterogeneity discussed in the rest of the review and applied to the interpretation of the studies. For example, in the section starting on page 6 on Risk of SPMs, the authors state that differences among the studies include exclusion vs. inclusion of TRb, the length of follow-up and treatment modalities. The authors provide information on the length of follow-up and inclusion/exclusion of TRb but not on time period which would provide information on treatment modalities. Also, there is no discussion about how those differences affect the results discussed. Based on these differences, which cohorts would be expected to have higher or lower incidence? The last paragraph of this section (starting line 236 on page 7) is excellent, guiding the reader in interpreting the results for heritable vs non-heritable survivors.

The section on Causes of SPM development and its subsections is generally well written and provides very useful interpretation and synthesis of the studies.

In the subsection Influence of genetic predisposition starting at line 309 on page 9, the authors highlight the difference in melanoma between patients with inherited vs de novo mutations. The 95% CIs for the cumulative incidence rates given overlap, meaning that the difference is unlikely to be statistically significant and may have occurred by chance. 

In the subsection Influence of targeted chemotherapy starting at line 434 on page 11, the authors refer to the Japanese survivor cohort and give the mean follow-up as 79 months but a few lines below say that the follow-up was almost 25 years. This seems inconsistent.

An additional sentence explaining why the reported TRb incidence could be less reliable from small cohorts would be helpful (line 457).

The two paragraphs in the Long-term surveillance section about TRb screening are not well organized.

Minor issues:

  1. Inconsistent terminology - Although heritable and non-heritable are most often used to describe rb, in places, the authors use hereditable/non-hereditable and hereditary/non-hereditary. The terminology should be consistent throughout.
  2. The abstract states that trilateral rb is a separate type of SPM, while the first paragraph of the body of the review states that TRb is not a SPM. Then there is section on page 12 specifically focused on TRb. The authors should state clearly in the abstract and first paragraph whether or not they consider TRb to be a SPM and whether the review will discuss it or not.
  3. In some places, references were not numbered, for example line 155.
  4. In the Multiple primary malignancies section starting at line 255 on page 7, the authors state that the risk of a next primary malignancy was increased 8-fold in patients who have had a first SPM. It would be good to state who those patients were being compared to - presumably those who did not have a first SPM. 
  5. On page 8 in the description of the Woo and Harbour analysis, it would be helpful to provide the years of treatment/diagnosis of the survivors included.

Reviewer 3 Report

This is an interesting review regarding trilateral retinoblastoma and malignancies following retinoblastoma. 

English needs to be improved, apart from single words,  some sentences throughout the text are not clear.

Following are many minor points to be considered and corrected:

  • The title of first paragraph “SPM definition and reason for susceptibility in heritable Rb population” is not necessary, can be removed.

  • The paragraph “Rb treatment over the past decades and factors influencing SPM reports” can be reduced to “Rb treatment over the past decades” including “Rb survival and therapies” and “Rb treatment in time and effects on SPM development”, while “Factors influencing SPM reports” should be a paragraph by itself.

  • Authors have by mistake left behind in the text indications for references as author name, journal and year instead of numbered references, see lines 102, 155, 457, 509, 513,528, 531.

  • Figure 1 legend is too long, needs to be shortened. Details on treatment changes over time are already present in the text.

  • Figure 2 legend is also too long. Important concepts are better to be placed in the text

  • The acronym SIR stands for “standard incidence ratio” (line 188) and not for “increased risk” (line 197)

  • line 201, the term “age-brackets” should be “age intervals” and in line 530 and line 531 “age interval”

  • The numbers 10,000 is incorrectly written in line 215 (10.000), line 220 and 222 (10 000)

  • In table 2 the information regarding the Danish cohort is too long and is alredy present in the text (lines 231-233)

  • In the text and figure 3 authors wrote “CNS” for “CNS tumor”, instead it should always be “CNS tumor”

  • In fgure 3 c is written “regios” instead of “region”

  • line 382 should be “alkylating chemotherapy drugs”, instead of “alkylating chemotherapy”

  • The whole paragraph “Characteristics and treatment of trilateral retinoblastoma” includes information reported in reference 3. It is better to write in the beginning of the paragraph that “DeJong et al [3] reported that...” instead of inserting four times the same reference number. Also treatment for TRb could be described in more dept.

  • line 482 I believe “this year” refers to 2020, it must be changed.

  • line 484 “brain and CNS tumors” should be “brain and other CNS tumors”

  • line 488-489 how is the “decreased visual activity” related to the recommendation for skin protection  education measures for Rb heritable survivors? Please clarify.

  • English must be improved, there are many sentences and words that need to be corrected, see for example:

line 11 should be “will be discussed” instead of “will discussed”

line 46-48 sentence  needs to be rephrased 

line 150 sentence is not clear

line 277-279 sentence is not clear

line 478 better “in fact” than “to illustrate”

line 499-500 in the sentence “no annual thyroid ultrasound should not be performed” should be changed to “annual thyroid ultrasound should not be performed “

Reviewer 4 Report

 I read with interest the review about the risk of the onset of other malignant neoplasms in patients with retinoblastoma.
Although the review is sometimes not very fluent in reading, I believe it may be worthy of publication.
I consider the title confusing, I would invite the authors to delete "primary"  from the title and the text "subsequent malignant neoplasms ..."

Round 2

Reviewer 1 Report

The authors sincerely responded to the review opinion.